# Evaluating Emotion Arcs Across Languages:
# Bridging the Global Divide in Sentiment Analysis

**Daniela Teodorescu**[1,2*] and **Saif M. Mohammad**[3]

[1]MaiNLP, Center for Information and Language Processing, LMU Munich, Germany
[2]Department of Computing Science, University of Alberta
[3]National Research Council Canada
daniela@cis.uni-muenchen.de, saif.mohammad@nrc-cnrc.gc.ca

## Abstract

Emotion arcs capture how an individual (or a population) feels over time. They are widely used in industry and research; however, there is little work on evaluating the automatically generated arcs. This is because of the difficulty of establishing the true (gold) emotion arc. Our work, for the first time, systematically and quantitatively evaluates automatically generated emotion arcs. We also compare two common ways of generating emotion arcs: Machine-Learning (ML) models and Lexicon-Only (LexO) methods. By running experiments on 18 diverse datasets in 9 languages, we show that despite being markedly poor at instance level emotion classification, LexO methods are highly accurate at generating emotion arcs when aggregating information from hundreds of instances. We also show, through experiments on six indigenous African languages, as well as Arabic, and Spanish, that *automatic translations* of English emotion lexicons can be used to generate high-quality emotion arcs in less-resource languages. This opens up avenues for work on emotions in languages from around the world; which is crucial for commerce, public policy, and health research in service of speakers often left behind. Code and resources:
https://github.com/dteodore/EmotionArcs

## 1 Introduction

Commercial applications as well as research projects often benefit from accurately tracking the emotions associated with an entity over time. Public health researchers are interested in analyzing social media posts to better understand population-level well-being (Vishnubhotla and Mohammad, 2022), loneliness (Guntuku et al., 2019), depression (De Choudhury et al., 2013), etc. Government Policy makers benefit from tracking public opinion over time for developing effective interventions and laws. For example, tracking sentiment towards

---

health interventions such as mask mandates and vaccine policies (Hu et al., 2021). Researchers in Digital Humanities are interested in understanding basic components of stories such as plot structures, how emotions are associated with compelling characters, categorizing stories based on emotion changes (Reagan et al., 2016), etc. In all of these applications, the goal is to determine whether the degree of a chosen emotion has remained steady, increased, or decreased from one time step to the next. The time steps of consideration may be days, weeks, years, etc. This series of time step–emotion value pairs, which can be represented as a time-series graph, is often referred to as an *emotion arc* (Mohammad, 2011; Reagan et al., 2016).

Automatic methods generate emotion arcs using:

- The text of interest where the individual sentences (or instances) are temporally ordered; possibly through available timestamps indicating when the instances were posted/uttered: e.g., all the tweets relevant to (or mentioning) a government policy along with their metadata that includes the date and time of posting.

- The emotion dimension/category of interest: e.g., anxiety, anger, valence, arousal, etc.

- The time step granularity of interest (e.g., days, weeks, months etc.)

The arcs are generated through a two-step process:

- Apply emotion labels to units of text. Two common approaches for labeling units of text are: (1) *The Lexicon-Only (LexO) Method:* to label words using emotion lexicons and (2) *The Machine-Learning (ML) Method:* to label whole sentences using supervised ML models (with or without using emotion lexicons).

- Aggregate the information to compute time step–emotion value scores; e.g., if using the lexicon approach: for each time step, compute the percentage of anger words in the target text pertaining to each time step, and if

---

*Work done while at the University of Alberta and internship with the National Research Council Canada.

using the ML approach: for each time step, compute the percentage of angry sentences for each time step.

Despite their wide-spread use in industry and research, there is little work on evaluating the generated emotion arcs (in part due to the difficulty of establishing the true (gold) emotion arc). This is problematic because we should know how accurate the generated emotion arcs are before drawing inferences from them or deploying these systems in applications. Further, different methods of generating emotion arcs (LexO or ML) have different characteristics. For example, LexO methods are interpretable, accessible, compute-friendly, and do not require annotated data (especially for each domain of application) (Mohammad, 2023, 2021; Öhman, 2021); whereas ML methods tend to consider context and longer-range dependencies, making them more accurate at labeling individual instances for emotions. (Table 5 in Appendix A depicts the pros and cons in more detail.) Thus, it is often assumed that ML methods are markedly more accurate than LexO methods for all tasks, including the creation of emotion arcs. However, this assumption may be false; and can only be tested for the task of generating emotion arcs when there is a mechanism to evaluate the arcs.

A robust and simple setup for evaluating emotion arcs also allows one to test different design decisions in how the arcs are generated. For example, we know little about how best to aggregate information when using emotion lexicons; e.g., is it better to use coarse or fine-grained lexicons, should we ignore slightly emotional words, etc.

Work on emotions has largely focused on English, excluding the voices, cultures, and perspectives of a majority of people from around the world. Such an exclusion is not only detrimental to those that are excluded, but also to the world as a whole. For example, western-centric science that only draws on data from western languages may falsely claim certain universals about language, emotions, and stories, that when tested on data from other cultures does not hold true anymore. Thus, working with more languages helps discover new paradigms that represent the world better.

However, generating emotion arcs in most languages other than English is stymied by the lack of emotion-labeled resources. Using translated labeled texts with ML methods is problematic (Mohammad et al., 2016; Hogenboom et al., 2014). However, for a number of NLP tasks, systems have benefited from using translations of lexical resources (from say English) into the target language. Thus, high-accuracy LexO methods for creating emotion arcs can open up avenues of work in various languages using (manual or automatic) translations of existing emotion lexicons.

Our work, for the first time, systematically and quantitatively evaluates automatically generated emotion arcs. We also compare the ML and LexO methods for generating arcs. Note, that our aim is not to build more accurate emotion classification systems, but rather we apply existing methods to evaluate emotion arcs generated across domains and languages. We conducted experiments on 18 datasets in nine languages from diverse domains, including: tweets, SMS messages, reviews, etc. (Two English datasets from movie reviews, four English datasets from SemEval 2014, six datasets from SemEval 2018 (two each in English, Arabic, and Spanish), and six African languages datasets from SemEval 2023 (AfriSenti).)

Emotions can be characterized by various dimensions and categories (e.g., valence, arousal, dominance vs. anger, fear, joy, sadness). In this work we explore the valence (or sentiment), and therefore generate valence (or sentiment) arcs.[1] Our work sets the foundation for future work which can extend analyses to various emotion categories.

## 2 Related Work

Emotion arcs have commonly been created from literary works and social media content. Alm and Sproat (2005) were the first to automatically classify sentences from literary works for emotions using a machine learning paradigm. Mohammad (2011) was the first to create emotion arcs and analyse the flow of emotions across the narrative in various novels and books using emotion lexicons. Kim et al. (2017) built on this work by creating emotion arcs to determine emotion information for various genres using the NRC Emotion Lexicon (Mohammad and Turney, 2013).[2] Reagan et al. (2016) and Del Vecchio et al. (2018) clustered emotion arcs and found evidence for six prototypical arc shapes in stories. Hipson and Mohammad (2021)

---

[1]**Terminology:** The positive–negative (or pleasure–displeasure) dimension has long been a focus of study, especially in psychology. They refer to this dimension as *valence*. However, early NLP work in the area made use of product and movie review datasets, and referred to the dimension as *sentiment* — a term that has stuck in subsequent NLP work.

[2]http://saifmohammad.com/WebPages/NRC-Emotion-Lexicon.htm

analysed emotion arcs for individual characters (instead of the whole narrative) in movie dialogues. Emotion arcs of literary works have been used to better understand plot and character development (Reagan et al., 2016; Kim et al., 2017; Hipson and Mohammad, 2021); and also for assisting writers develop and improve stories (Ashida et al., 2021; Somasundaran et al., 2020).

Recently, Hipson and Mohammad (2021) introduced how the patterns with which emotions change over time — emotion dynamics — can be inferred from text. Vishnubhotla and Mohammad (2022) computed UEDs from tweets and analysed how they have changed over the years. Teodorescu et al. (2023b) studied emotional development in children's writing and found meaningful patterns across age. In the public health domain, Teodorescu et al. (2023a) created emotion arcs for tweeters who self-disclosed as having a mental health diagnosis (on Twitter). They found that certain emotion dynamics patterns differed significantly between tweeters with mental health conditions (e.g., attention deficit hyperactivity disorder, depression, bipolar) compared to the control group.

However, there is surprisingly little work on arc evaluation. A key reason for this is that it is hard to determine the true emotion arc of a story from data annotation. One attempt to evaluate aspects of an emotion arc can be seen in Bhyravajjula et al. (2022). They asked one volunteer to read mini-segments of a *'The Lord of the Rings'* novel to determine whether the protagonist's circumstance undergoes a positive or negative shift. They then determined the extent to which the automatic method captured the same shifts. Here we propose a simpler alternative way to robustly evaluate automatically generated emotion arcs on a wide variety of domains and multiple languages.

## 3 Experimental Setup to Evaluate Automatically Generated Emotion Arcs

We begin by describing the evaluation setup. This is a key contribution of this work since no prior work systematically evaluates automatically generated emotion arcs. We construct *gold emotion arcs* from existing datasets where individual instances are manually annotated for valence (sentiment). Here, an instance could be a tweet, a sentence from a customer review, a sentence from a personal blog, etc. Depending on the dataset, manual annotations for an instance may represent the emotion of a speaker, or sentiment towards a product or entity. For a pre-chosen bin size of say 100 instances per bin, we compute the gold emotion score by taking the average of the human-labeled emotion scores of the instances in that bin (in-line with the commerce and social media use cases discussed earlier). We move the window forward by one instance, compute the average in that bin, and so on. (Using larger window sizes does not impact conclusions.) We created text streams for the experiments in Section 4 by ordering instances from a dataset by increasing gold score before binning. This created arcs with a more consistent emotion change rate and we first explore results in this scenario. In Section 7, we look at the impact of more dynamic emotion changes (e.g., varying peak heights and widths) on performance.

We automatically generate emotion arcs using the LexO and ML methods discussed in the two sections ahead. We standardized all arcs (aka z-score normalization) so that the gold arcs are comparable to automatically generated arcs.[3] Finally, we evaluate the closeness of automatically generated emotion arcs with gold arcs using two metrics: linear correlation and Root Mean Squared Error (RMSE). We use Spearman rank correlation (Spearman, 1987) (range: -1 to 1). High correlation implies greater fidelity: no matter what the overall shape of the gold emotion arc, when it goes up (or down), the predicted emotion arc also goes up (or down).[4] RMSE (range: 0–∞) is a measure of the error between the true and predicted values. It penalizes bigger errors more, and thus is sensitive to outliers. Scores closer to 0 indicate better predictions. In our experiments, the correlation and RMSE scores are determined for thousands of points between the predicted and gold arcs.[5]

Table 1 shows key details of the English instance-labeled datasets. To determine whether using automatic translations of English lexicons into relatively less-resource languages is a viable option, we also experiment with instance-labeled datasets in Arabic (Ar), Spanish (Es), Amharic (Am), Hausa (Ha), Igbo (Ig), Kinyarwanda (Kr), Swahili (Sw), and Yoruba (Yo). Just as the English set, these contain original tweets (not translated) with emotion

---

[3]Subtract the mean from the score and divide by the standard deviation (arcs then have zero mean and unit variance).

[4]Different orderings of a given dataset produce different shapes; however, the Spearman correlation between the automatic and gold arcs for that dataset remains the same.

[5]For example, in a dataset with 3K instances, when using a rolling window of bin size 300, the arc consists of 2700 points.

| Dataset | Source | Domain | Dimension | Label Type | # Instances |
|---------|--------|--------|-----------|-----------|-------------|
| Movie Reviews Categorical | Socher et al. (2013) | movie | valence | categorical (0, 1) | 11,272 |
| Movie Reviews Continuous | Socher et al. (2013) | reviews | valence | continuous (0 to 1) | 11,272 |
| SemEval 2014 | Rosenthal et al. (2014) | multiple* | valence | categorical (-1, 0, 1) | multiple* |
| SemEval 2018 (V-OC) | Mohammad et al. (2018) | tweets | valence | categorical (-3,-2,...3) | 2,567 |
| SemEval 2018 (V-Reg) | Mohammad et al. (2018) | tweets | valence | continuous (0 to 1) | 2,567 |

Table 1: Key details of the **English emotion-labeled datasets** used. *The SemEval 2014 dataset is a collection of 18,611 instances, including LiveJournal posts (1141), SMS messages (2082), tweets (15302), sarcastic tweets (86).

| Lexicon | Source | Categories / Dimensions | Label Type | # Terms |
|---------|--------|------------------------|-----------|---------|
| NRC VAD | Mohammad (2018) | **valence**, arousal, dominance | continuous (-1 to 1) | 20,007 |
| | | **valence**, arousal, dominance | categorical (-1, 0, 1) | 20,007 |

Table 2: Lexicons used in this study. The subset of emotions explored in our experiments are marked in bold.

labels by native speakers. The datasets are of two kinds: those with categorical labels such as the SemEval 2014, which has -1 (negative), 0 (neutral), and 1 (positive), as well as those with continuous labels such as SemEval 2018 (V-Reg), which has real-valued sentiment intensity scores between 0 (lowest/no intensity) and 1 (highest intensity).

In all, we conducted experiments with 18 emotion-labeled datasets from nine languages, with labels that are either categorical or continuous, for valence. The results also establish key benchmarks; useful to practitioners for estimating the quality of the arcs under various settings.

## 4 LexO Arcs: Emotion Arcs Generated from Counting Emotion Words

Recall that in the Introduction we described how emotion arcs are automatically generated. Key parameters in that process include bin size, type of emotion lexicon used (e.g., categorical or continuous emotion scores), and how to handle terms in the text that are not in the lexicon (OOV terms).

Choice of bin size depends on the application and available data. For example, if a company wants to know how the proportion of angry comments has changed from month to month in the last 10 years, then the bin size to use is month. If instead, they want to know how things changed on a day-by-day basis over the last 45 days, then the bin size to use is a day. Depending on the volume of relevant data and the bin size chosen, the bins may include a higher or lower number of instances. For our experiments we explored various bin sizes (1, 10, 30, 50, 100, 200, and 300).

Table 2 shows the English emotion lexicon we used: The NRC VAD Lexicon.[6] It has both categorical and real-valued versions, which is useful to study whether using fine-grained lexicons leads to

[6] http://saifmohammad.com/WebPages/nrc-vad.html

markedly better emotion arcs.

The two OOV handling methods explored were: 1. Assign label NA (no score available) and disregard these words, and 2. Assign 0 score (neutral or not associated with emotion category), thereby, leading to a lower average score for the instance than if the word was disregarded completely.

We then evaluated how closely the arcs correspond to the gold valence arcs. In Appendix B, we show the impact of using emotion lexicons more selectively—using only the entries that have an emotion association greater than some predetermined threshold.

**Results (Valence):** Figure 1 shows the correlations and RMSE values between predicted valence arcs and the gold valence arcs.

*Bin Size:* Overall, across datasets and regardless of the type of lexicon used and how OOV words are handled, increasing the bin size dramatically improves correlation with the gold arcs. In fact, with bin sizes as small as 50, many of the generated arcs have correlations above 0.9. With bin size 100 and above correlations approach high 0.90's. Similarly, RMSE starts at approximately 1.20 to 1.10 at bin size 1, and quickly drops as bin size increases. At bin size 300, RMSE approaches quite low scores (0.3–0.1).

If we use a bin size of 1, we only get the ups and downs of the emotions associated with individual words, which will likely be quite far off from the true emotion arc. As bin size increases (more data per bin), the predicted arc gets closer to the true arc probably because:

> If bin $x$ is expressing more of an emotion (higher emotion intensity) than bin $y$, then $x$ has a higher average word–emotion association score than $y$.

*Categorical vs. Real-Valued Lexicons:* Using a real-valued lexicon obtains higher correlations across bin sizes, methods for processing OOV terms, and

| Test Data | Test Data Language | Emotion | OOV | Lexicon Scores | Bin Size | | | | | | Bin Size | | | | | |
|---|---|---|---|---|---|---|---|---|---|---|---|---|---|---|---|---|
| | | | | | 1 | 10 | 30 | 50 | 100 | 300 | 1 | 10 | 30 | 50 | 100 | 300 |
| Movie Reviews Categorical | Eng | valence | assigned NA | categorical | 0.286 | 0.665 | 0.817 | 0.848 | 0.862 | 0.863 | 1.20 | 0.84 | 0.62 | 0.53 | 0.42 | 0.28 |
| | | | | real-valued | 0.329 | 0.724 | 0.842 | 0.859 | 0.864 | 0.866 | 1.17 | 0.78 | 0.56 | 0.47 | 0.38 | 0.25 |
| | | | assigned neutral score | categorical | 0.296 | 0.701 | 0.829 | 0.852 | 0.861 | 0.862 | 1.18 | 0.81 | 0.61 | 0.54 | 0.45 | 0.31 |
| | | | | real-valued | 0.335 | 0.750 | 0.847 | 0.859 | 0.863 | 0.864 | 1.15 | 0.76 | 0.56 | 0.50 | 0.42 | 0.30 |
| Movie Reviews Continuous | Eng | valence | assigned NA | categorical | 0.326 | 0.733 | 0.881 | 0.917 | 0.951 | 0.980 | 1.16 | 0.74 | 0.51 | 0.42 | 0.32 | 0.20 |
| | | | | real-valued | 0.374 | 0.786 | 0.912 | 0.941 | 0.965 | 0.986 | 1.12 | 0.67 | 0.44 | 0.36 | 0.27 | 0.16 |
| | | | assigned neutral score | categorical | 0.344 | 0.763 | 0.890 | 0.921 | 0.951 | 0.978 | 1.14 | 0.70 | 0.48 | 0.40 | 0.32 | 0.23 |
| | | | | real-valued | 0.386 | 0.805 | 0.915 | 0.942 | 0.965 | 0.985 | 1.11 | 0.64 | 0.43 | 0.35 | 0.28 | 0.20 |
| SemEval 2014 (LiveJournal) | Eng | valence | assigned NA | categorical | 0.333 | 0.742 | 0.872 | 0.900 | 0.932 | 0.986 | 1.16 | 0.75 | 0.51 | 0.41 | 0.28 | 0.15 |
| | | | | real-valued | 0.404 | 0.820 | 0.907 | 0.925 | 0.939 | 0.981 | 1.09 | 0.65 | 0.44 | 0.36 | 0.27 | 0.16 |
| | | | assigned neutral score | categorical | 0.317 | 0.725 | 0.842 | 0.878 | 0.924 | 0.987 | 1.18 | 0.77 | 0.55 | 0.46 | 0.34 | 0.22 |
| | | | | real-valued | 0.378 | 0.811 | 0.896 | 0.923 | 0.939 | 0.988 | 1.11 | 0.65 | 0.44 | 0.37 | 0.28 | 0.20 |
| SemEval 2014 (SMS) | Eng | valence | assigned NA | categorical | 0.265 | 0.609 | 0.768 | 0.810 | 0.862 | 0.915 | 1.22 | 0.87 | 0.64 | 0.55 | 0.42 | 0.26 |
| | | | | real-valued | 0.343 | 0.716 | 0.853 | 0.879 | 0.895 | 0.930 | 1.15 | 0.72 | 0.47 | 0.38 | 0.28 | 0.16 |
| | | | assigned neutral score | categorical | 0.274 | 0.656 | 0.819 | 0.864 | 0.882 | 0.907 | 1.21 | 0.82 | 0.56 | 0.44 | 0.30 | 0.17 |
| | | | | real-valued | 0.337 | 0.738 | 0.862 | 0.885 | 0.894 | 0.916 | 1.15 | 0.71 | 0.44 | 0.31 | 0.18 | 0.09 |
| SemEval 2014 (tweets sarcasm) | Eng | valence | assigned NA | categorical | 0.175 | 0.738 | 0.948 | 0.927 | | | 1.26 | 0.68 | 0.23 | 0.33 | | |
| | | | | real-valued | 0.234 | 0.814 | 0.954 | 0.952 | | | 1.21 | 0.54 | 0.18 | 0.23 | | |
| | | | assigned neutral score | categorical | 0.206 | 0.587 | 0.830 | 0.873 | | | 1.28 | 0.85 | 0.43 | 0.48 | | |
| | | | | real-valued | 0.270 | 0.722 | 0.830 | 0.935 | | | 1.24 | 0.69 | 0.39 | 0.36 | | |
| SemEval 2014 (tweets) | Eng | valence | assigned NA | categorical | 0.231 | 0.572 | 0.753 | 0.819 | 0.877 | 0.907 | 1.24 | 0.90 | 0.66 | 0.56 | 0.44 | 0.32 |
| | | | | real-valued | 0.323 | 0.711 | 0.854 | 0.891 | 0.910 | 0.917 | 1.16 | 0.73 | 0.50 | 0.42 | 0.32 | 0.24 |
| | | | assigned neutral score | categorical | 0.244 | 0.609 | 0.789 | 0.851 | 0.897 | 0.913 | 1.23 | 0.88 | 0.64 | 0.53 | 0.42 | 0.30 |
| | | | | real-valued | 0.322 | 0.728 | 0.868 | 0.898 | 0.912 | 0.915 | 1.16 | 0.73 | 0.50 | 0.40 | 0.30 | 0.22 |
| SemEval 2018 (V-OC) | Eng | valence | assigned NA | categorical | 0.448 | 0.851 | 0.949 | 0.964 | 0.972 | 0.986 | 1.06 | 0.57 | 0.34 | 0.27 | 0.21 | 0.15 |
| | | | | real-valued | 0.502 | 0.879 | 0.958 | 0.970 | 0.978 | 0.987 | 1.01 | 0.51 | 0.29 | 0.23 | 0.17 | 0.12 |
| | | | assigned neutral score | categorical | 0.427 | 0.850 | 0.944 | 0.964 | 0.976 | 0.980 | 1.08 | 0.58 | 0.36 | 0.28 | 0.22 | 0.16 |
| | | | | real-valued | 0.476 | 0.880 | 0.954 | 0.970 | 0.979 | 0.984 | 1.03 | 0.52 | 0.32 | 0.25 | 0.18 | 0.13 |
| SemEval 2018 (V-Reg) | Eng | valence | assigned NA | categorical | 0.458 | 0.839 | 0.936 | 0.956 | 0.978 | 0.994 | 1.05 | 0.59 | 0.38 | 0.31 | 0.22 | 0.13 |
| | | | | real-valued | 0.513 | 0.867 | 0.945 | 0.963 | 0.980 | 0.997 | 1.00 | 0.53 | 0.34 | 0.28 | 0.20 | 0.13 |
| | | | assigned neutral score | categorical | 0.438 | 0.845 | 0.940 | 0.959 | 0.978 | 0.997 | 1.07 | 0.59 | 0.37 | 0.30 | 0.22 | 0.12 |
| | | | | real-valued | 0.488 | 0.874 | 0.948 | 0.964 | 0.979 | 0.997 | 1.02 | 0.54 | 0.34 | 0.28 | 0.20 | 0.11 |

Corr 0.000 — 1.000    RMSE 0.09 — 1.28

Figure 1: **Valence Arcs:** Spearman correlation and RMSE values between LexO and gold arcs of **English datasets**.

datasets. The difference is marked for very small bin sizes (such as 1 and 10) but progressively smaller for higher bin sizes. Entries from real-valued lexicons carry more fine-grained emotion information, and it is likely that this extra information is especially helpful when arcs are determined from very little text (as in the case of small bins).

*OOV Terms:* Results with the two OOV-handling methods were comparable (no clear winner).

**Discussion:** For many social media applications, one has access to tens of thousands, if not millions of posts. There, it is not uncommon to have time-steps (bins) that include thousands of instances. Thus, it is remarkable that even with relatively small bin sizes of a few hundred, the simple lexicon approach is able to obtain very high correlations. Of course, the point is not that the lexicon approach is somehow special, but rather that aggregation of information can very quickly generate high quality arcs, even if the constituent individual emotion signals are somewhat weak.

## 5 ML Arcs: Emotion Arcs Generated from Counting ML-Labeled Sentences

This section explores how the accuracy of instance-level (sentence- or tweet-level) emotion labeling impacts the quality of the generated emotion arcs. We approached this by creating an 'oracle' system, which has access to the gold instance emotion labels.

There are several metrics for evaluating sentiment analysis at the instance level such as accuracy, correlation, or F-score. However, we focus on accuracy as it is a simple intuitive metric. Inputs to the Oracle system are a dataset of text (for emotion labelling) and a level of accuracy (e.g., 90% accuracy) to perform instance-level emotion labelling at. Then, the system goes through each instance and predicts the correct emotion label with a probability corresponding to the pre-chosen accuracy. In the case where the system decides to assign an incorrect label, it chooses one of the possible incorrect labels at random.

| Test data | Emotion | Instance OS: Accuracy | Bin Size | | | | | |
|---|---|---|---|---|---|---|---|---|
| | | | 1 | 10 | 50 | 100 | 200 | 300 |
| SemEval 2018 (V-OC) | valence | 0 | -0.125 | -0.390 | -0.703 | -0.801 | -0.872 | -0.893 |
| | | 20 | 0.092 | 0.262 | 0.489 | 0.607 | 0.783 | 0.846 |
| | | 40 | 0.320 | 0.715 | 0.927 | 0.959 | 0.979 | 0.985 |
| | | 60 | 0.505 | 0.861 | 0.965 | 0.977 | 0.986 | 0.992 |
| | | 80 | 0.730 | 0.963 | 0.982 | 0.984 | 0.988 | 0.995 |
| | | 100 | 1.000 | 1.000 | 1.000 | 1.000 | 1.000 | 1.000 |

Figure 2: **Valence Arcs:** Spearman correlations of the arcs generated using the **valence-classification Oracle System (OS)** with the gold arcs of English tweets. OS (Accuracy): Accuracy at instance-level sentiment classification.

| Source | Model | Instance-Level Accuracy | Bin Size | | | | | |
|---|---|---|---|---|---|---|---|---|
| | | | 1 | 10 | 50 | 100 | 200 | 300 |
| Socher et al. (2013) | RNTN | 85.4% | 0.829 | 0.972 | 0.980 | 0.983 | 0.986 | 0.992 |
| Devlin et al. (2019) | BERT-base | 93.5% | 0.921 | 0.981 | 0.984 | 0.988 | 0.993 | 0.996 |
| Devlin et al. (2019) | BERT-large | 94.9% | 0.932 | 0.986 | 0.984 | 0.988 | 0.993 | 0.996 |
| Yang et al. (2019) | XLNet | 97.0% | 0.959 | 0.990 | 0.985 | 0.988 | 0.993 | 0.996 |
| Liu et al. (2019) | RoBERTa | 96.4% | 0.958 | 0.989 | 0.986 | 0.989 | 0.994 | 0.997 |

Table 3: **Valence Arcs:** Spearman correlations between arcs generated **using neural models** for instance-level sentiment classification and gold arcs of English tweet streams.

We use the Oracle to generate emotion labels pertaining to various levels of accuracy, for the same dataset. We then generate emotion arcs just as described in the previous section (by taking the average of the scores for each instance in a bin), and evaluate the generated arcs just as before (by determining correlation with the gold arc). This Oracle System allows us to see how accurate an instance level emotion labelling approach needs to be to obtain various levels of quality when generating emotion arcs.

Figure 2 shows the correlations of the valence arcs generated using the Oracle System with the gold valence arcs created from the SemEval 2018 V-OC test set (that has 7 possible labels: -3 to 3).[7] We observe that, as expected the Oracle Systems with instance-level accuracy greater than approximately the random baseline (14.3% for this dataset) obtain positive correlations; whereas those with less than 14% accuracy obtain negative correlations. As seen with the results of the previous section, correlations increase markedly with increase in bin size. Even with an instance-level accuracy of 60%, correlation approaches 1 at larger bin sizes. Overall, we again observe high quality emotion arcs with bin sizes of just a few hundred instances.

Table 3 shows the correlations obtained on the same dataset when using various deep neural network and transformer-based systems. Observe that the recent systems obtain nearly perfect correlation at bin sizes 200 and 300. However, for a given application scenario, these results can only be obtained when the machine learning system is able to train on sufficient domain-specific training data (which is often scarce), the computer power, and knowledge to work with these systems is accessible. For applications where simple, interpretable, low-cost, and low-carbon-footprint systems are desired, the lexicon-based systems described in the previous section, are often more suitable.

## 6 LexO Arcs: Emotion Arcs Generated from Translated Emotion Lexicons

Given the competitive performance of the Lexicon-Only (LexO) method, and the many benefits of the LexO method such as their simplicity and interpretability, we now explore whether high-quality emotion arcs can be created for low-resourced languages using automatic translations of English emotion lexicons. Specifically, we make use of translations of the NRC lexicons from English into the language of interest and perform similar experiments as described in the previous section.[8] We explore bin sizes of 400 and 500 as well, as we expect that instance-level accuracy will be lower in the translated-lexicon case.

We would like to note that automatic translations may not be available for many very low resource languages. Currently, Google Translate has functionality to translate across about 120 languages.

---

[7] Figure 5 (Appendix) shows similar Oracle System results for other datasets.

[8] The NRC Emotion and VAD Lexicon packages come with translations into over 100 languages.

| Language | Source | Domain | Dimension | Label Type | # Instances |
|---|---|---|---|---|---|
| Amharic | Yimam et al. (2020) | tweets | valence | categorical (-1, 0, 1) | 5,984 |
| Hausa | Muhammad et al. (2022) | tweets | valence | categorical (-1, 0, 1) | 14,172 |
| Igbo | Muhammad et al. (2022) | tweets | valence | categorical (-1, 0, 1) | 10,192 |
| Kinyarwanda | Muhammad et al. (2023) | tweets | valence | categorical (-1, 0, 1) | 3,302 |
| Swahili | Muhammad et al. (2023) | tweets | valence | categorical (-1, 0, 1) | 1,810 |
| Yoruba | Muhammad et al. (2022) | tweets | valence | categorical (-1, 0, 1) | 8,522 |

Table 4: Key information pertaining to the **African language emotion-labeled tweets datasets** we use in our experiments. The No. of instances includes the train set for the SemEval 2023 Task 12.

This may seem large (and it is indeed a massive improvement from just a decade back), but thousands of languages still remain on the other side of the digital divide. Thus, even requiring word translations is a significant limitation for many languages.

We generated and evaluated emotion arcs in six African languages: Amharic, Hausa, Igbo, Kinyarwanda, Swahili, and Yoruba (using the datasets shown in Table 4). These are some of the most widely spoken indigenous African languages.[9] Table 6 in the Appendix presents details about each, including their language family and the primary regions where they are spoken. Note that these languages themselves are diverse covering three different language families: Afroasiatic, Bantu, and Niger-Congo. Swahili, is more influenced by the Indo-European languages than Hausa —about 40% of its vocabulary is made up of Arabic loan words. We contrast results on these languages to Arabic, a commonly spoken language in Northern Africa which has somewhat more resources than indigenous African languages. Arabic still has much fewer NLP resources than English. We also contrast our results to Spanish, which has fewer resources than English; and is more similar to English than Arabic. (The Ar and Es valence-labeled datasets used are shown in Table 7 in the Appendix.) High fidelity of the predicted arcs with the gold arcs will unlock the potential for research in affect-related psychology, health science, and digital humanities for a vast number of languages.

## 6.1 Valence Arcs: Using Translated Lexicons

We ran a wide range of experiments in Arabic, Spanish, and the African languages with all the parameter variations discussed earlier. We found the same trends for the OOV handling and lexicon granularity parameters for these languages as for English. Therefore, for brevity, Figure 3 only shows the results with using real-valued lexicons (which performed better than categorical score lexicons)

and assign label NA to OOV words (which was comparable to the 'assign neutral score' method). Figure 3 shows the results for generating emotion arcs using translations of English lexicons into Arabic, Spanish, and the African languages for both categorically (e.g., V-OC and SemEval 2023) and continuously (e.g., V-Reg) labelled valence datasets. We also provide the English results on the corresponding datasets for easy comparison.

For Arabic and Spanish (V-OC and V-Reg datasets): Correlations start at 0.40–0.50 at the instance-level (bin size 1), and reach 0.98–0.99 from bin sizes 200 and onwards. Performance using translations of English lexicons does surprisingly well. At bin size 1 English does performs better by about 10 points, however this difference quickly dissipates at bin size 50 and onwards.

For the African language texts (the SemEval 2023 datasets): Correlations start at lower values at the instance-level, than for Arabic and Spanish. At bin size 1, correlations are about 0.1–0.2 with the exception of Hausa at 0.374. As seen previously, with increasing bin size we are able to gain substantial improvements in performance by aggregating information. With a bin size of 200 and onwards correlations reach mid 0.90's which is a large gain in performance compared to instance-level. Among the African languages, at bin size 1–50 arcs for Hausa are by far the best, followed by arcs for Amharic; then arcs for Igbo, Kinyarwanda, and Swahili perform similarly; followed by Yoruba. For bin sizes 100 and onwards, Hausa and Kinyarwanda become the best performing, followed by Igbo and Amharic performing similarly, followed by Yoruba.

*Discussion:* Using translations of English lexicons allows us to create high-quality emotion arcs in African languages. The variation in performance across the African languages could be because of the variation in quality of automatic translations of the emotion lexicons across languages and different inter-annotator agreements for the instance-level emotion labels in the different datasets.

[9] https://www.pangea.global/blog/2018/07/19/10-most-popular-african-languages/

| Test Data | Test Lang. | Emotion | Test Type | Bin | | | | | | | |
|---|---|---|---|---|---|---|---|---|---|---|---|
| | | | | 1 | 10 | 50 | 100 | 200 | 300 | 400 | 500 |
| SemEval 2018 (V-OC) | Ar | Valence | categorical | 0.453 | 0.828 | 0.942 | 0.957 | 0.969 | 0.979 | 0.996 | 0.997 |
| | Eng | Valence | categorical | 0.502 | 0.879 | 0.970 | 0.978 | 0.984 | 0.987 | 0.991 | 0.998 |
| | Es | Valence | categorical | 0.419 | 0.802 | 0.939 | 0.970 | 0.982 | 0.983 | 0.992 | 0.999 |
| SemEval 2018 (V-Reg) | Ar | Valence | continuous | 0.457 | 0.806 | 0.939 | 0.969 | 0.991 | 0.998 | 0.996 | 0.999 |
| | Eng | Valence | continuous | 0.513 | 0.867 | 0.963 | 0.980 | 0.991 | 0.997 | 0.999 | 0.999 |
| | Es | Valence | continuous | 0.419 | 0.814 | 0.960 | 0.978 | 0.990 | 0.996 | 0.999 | 0.999 |
| SemEval 2023 (AfriSenti) | am | Valence | categorical | 0.259 | 0.638 | 0.878 | 0.912 | 0.918 | 0.920 | 0.927 | 0.935 |
| | ha | Valence | categorical | 0.374 | 0.728 | 0.851 | 0.896 | 0.928 | 0.936 | 0.940 | 0.943 |
| | ig | Valence | categorical | 0.209 | 0.561 | 0.819 | 0.880 | 0.913 | 0.923 | 0.930 | 0.934 |
| | kr | Valence | categorical | 0.231 | 0.569 | 0.842 | 0.911 | 0.940 | 0.943 | 0.946 | 0.953 |
| | sw | Valence | categorical | 0.231 | 0.576 | 0.835 | 0.855 | 0.868 | 0.871 | 0.888 | 0.893 |
| | yo | Valence | categorical | 0.118 | 0.238 | 0.419 | 0.497 | 0.601 | 0.647 | 0.671 | 0.704 |

Figure 3: **Valence Arcs:** Spearman correlations between arcs generated using **translated emotion lexicons** and gold arcs created from *categorically* and *continuously* labeled test data in Arabic, Spanish, and the African languages. The untranslated English results are shown for comparison.

Although the results for Arabic and the African languages are on different datasets, it is of interest to contrast the performance of Arabic to the African languages because Google's translations of words in the lexicon to Arabic is expected to be of higher quality than to the African languages (due to the relatively higher amounts of English–Arabic parallel text). Overall, it is interesting to note that the translation method creates relatively better arcs when the target language is closer to the source language (e.g., English to Spanish). However, even for distant languages, performance can be improved by increasing bin size (e.g., some African languages). Of course, increasing bin size means lower granularity, but in many application scenarios, and for drawing broad trends, it makes sense to aggregate hundreds or thousands of instances in each bin, leading to more reliable inferences about emotion trends over time.

## 7 Arc Quality under Varying Emotion Amplitudes and Spikiness

Aspects of how emotions change across the bins also impacts the quality of generated arcs. For example, for a given surge or dip of emotion, if the change of emotion strength is too small in magnitude (*small amplitude*) or occurs for too short of a time period (*high spikiness*), then the automatic method may fail to register the surge/dip. We will refer to arcs with many such emotion changes as *more dynamic* than those with more gradual and longer-duration emotion changes.

Recall that in Section 4 we used text streams ordered by increasing gold score. Now, to test the robustness of the LexO method, when dealing with dynamic emotion changes, we created new text streams by reordering the tweets in the emotion-labeled datasets in more random and dramatic ways. We created these dynamic text streams by sampling tweets from the chosen dataset with replacement until 200 crests and 200 troughs of various amplitudes and spikiness were obtained.[10] The gold arc is then standardized as before. We will refer to this new text stream generated from a dataset as *[dataset_name]-dynamic*, and the gold arcs created with this process as *dynamic gold arcs*.

The gold line in Figure 4 shows the beginning portion of the gold valence arc produced from the SemEval 2018 (V-Reg) dataset (bin size = 100). Observe that some peaks are rather small in amplitude whereas some others are greater than two standard deviations. Also, some emotion changes occur across a wider span of data (x-axis) whereas some others show large emotion change and back in a short span.

We then generated the predicted arc for this dataset using the LexO method, the NRC VAD lexicon, and by ignoring terms not found in the lexicon when determining bin scores. The green line in Figure 4 shows the beginning portion of the gold valence arc when using a rolling window with bin size 100. Observe that the green line traces the gold line quite closely. There are also occasions where it overshoots or fails to reach the gold arc. Figure 4 shows gold and predicted arcs for the Hausa dataset (bin size = 300). Observe that here

---

[10]Sampling with replacement allowed reuse of the limited labeled data to produce a long wave.

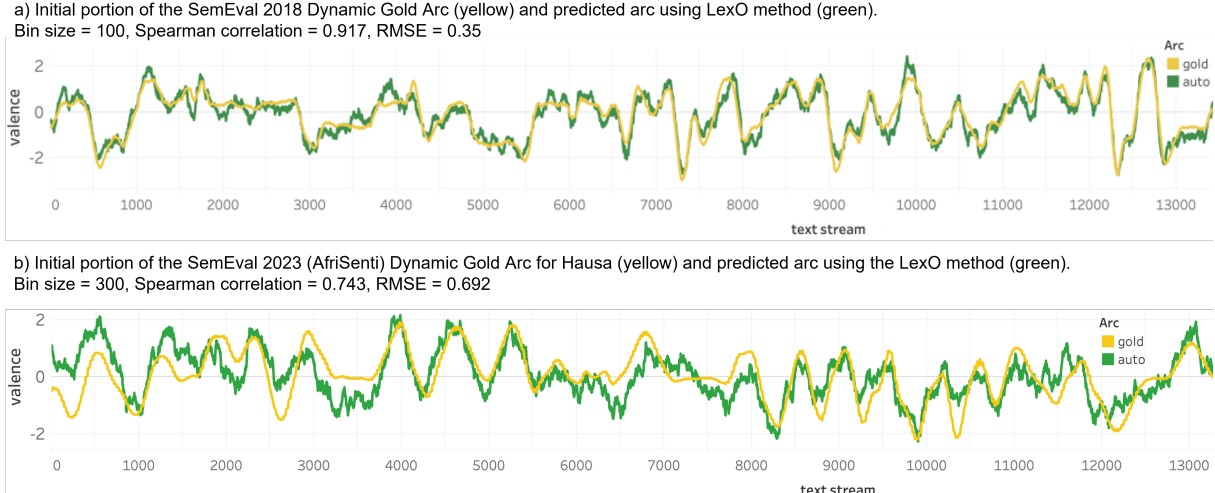

a) Initial portion of the SemEval 2018 Dynamic Gold Arc (yellow) and predicted arc using LexO method (green). Bin size = 100, Spearman correlation = 0.917, RMSE = 0.35

b) Initial portion of the SemEval 2023 (AfriSenti) Dynamic Gold Arc for Hausa (yellow) and predicted arc using the LexO method (green). Bin size = 300, Spearman correlation = 0.743, RMSE = 0.692

Figure 4: Dynamic and predicted arcs for English (a) and Hausa (b) using the LexO method.

the predicted arc still follows the gold arc closely (but as expected, not as closely as a.)

In experiments modifying bin size, the predicted arcs obtain a high correlation (above 0.9) with a bin size of 100 (and above), even for the dynamic text stream. However, these scores are (as expected) consistently lower than what were observed for the corresponding non-dynamic text stream. Note that the difference in scores is more pronounced for the smaller bin sizes than for larger ones. Therefore, in scenarios with smaller amounts of data per bin, judicious use of parameters described in Section 4 (e.g., treating OOVs as neutrals, using real-valued lexicons, etc.) can have marked influence on arc quality. Overall, though, these results show that the LexO method performs rather well even in the case of highly dynamic emotion changes.

## 8 Concluding Remarks

This work made contributions in two broad directions: First, we showed how methods for predicting emotion arcs can be evaluated, and used it to systematically and quantitatively evaluate both Lexicon-Only and Machine Learning methods. Second, we showed that using translations of English lexicons into the language of interest can generate high quality emotion arcs, even for languages that are very different from English.

Both ML and LexO methods produced high-quality emotion arcs when using bin size 50 and above (e.g., with bin size 100: obtaining correlations with gold valence arcs exceeding 0.98). ML methods are able to obtain markedly higher correlations at very low bin sizes (<50). However,

with the abundance of textual data available from social media, for many applications where one is aggregating hundreds or thousands of instances in each bin, the gains of ML methods in terms of correlation scores are miniscule.

With the cross-lingual experiments, we created emotion arcs from texts in six indigenous African languages, as well as Arabic and Spanish. For all of them, emotion arcs obtained high correlations as the bin size (aggregation level) was set to at least a few hundred instances. Correlations between predicted and gold emotion arcs were in general higher when the target language was closer to English.

In the last part of the paper, we explored how depending on the data available and the desired granularity of emotion arcs interacts with performance for various arc *amplitudes* and arc *spikiness*. We showed that the LexO method performs rather well even in the case of highly dynamic emotion changes. However, for smaller bin sizes, the predicted arc may markedly overshoot or not fully capture the gold peak or trough.

In all, we conducted experiments with 18 sentiment-labeled datasets from 9 languages, Thus the conclusions drawn are rather robust. The results on individual datasets also establish key benchmarks; useful to practitioners for estimating the quality of the arcs under various settings. Finally, since this work shows that simple lexicon-only approaches produce accurate arcs, practitioners from all fields, those working in low-resource languages, those interested in interpretability, or those without the resources to deploy neural models, can easily generate high-quality emotion arcs for their data.

## 9 Limitations

It is challenging to determine the true emotion arc of a text stream or story through data annotation. This is because usually data annotation is performed on smaller pieces of text such as sentences or paragraphs. It is difficult for people to read a large body of text, say from a novel, and produce consistent annotations for how emotions have changed as the text has progressed from start to finish. In the context of tweets, taking the average emotion scores of individual sentences is a reasonable option (as we have done here); however, that may not capture true emotions when dealing with large pieces of text written by the same person, for example, emotion arcs of stories in a novel. We hope future work will determine how to create gold arcs in such cases, and how different such arcs are from the arcs created by averaging the scores of individual sentences.

Constructing emotion lexicons from human annotations of emotion scores takes time and resources. Thus, these resources exist only for a handful of languages. As a feasible approach in the meanwhile, we uses automatic translations of emotion lexicons from English to a desired target language to generate emotion arcs in various languages. While this approach produces highly accurate emotion arcs, there are several considerations: different languages and cultures express emotions differently, there may be lexical gaps among languages, and characteristics and meaning can be lost in the translation. Therefore, one would expect that using human annotated lexicons would lead to higher emotion arc quality.

## 10 Ethics

Our research interest is to study emotions at an aggregate/group level. This has applications in determining public policy (e.g., pandemic-response initiatives) and commerce (understanding attitudes towards products). However, emotions are complex, private, and central to an individual's experience. Additionally, each individual expresses emotion differently through language, which results in large amounts of variation. Therefore, several ethical considerations should be accounted for when performing any textual analysis of emotions (Mohammad, 2022, 2020). The ones we would particularly like to highlight are listed below:

- Our work on studying emotion word usage should not be construed as detecting how people feel; rather, we draw inferences on the emotions that are conveyed by users via the language that they use.

- The language used in an utterance may convey information about the emotional state (or perceived emotional state) of the speaker, listener, or someone mentioned in the utterance. However, it is not sufficient for accurately determining any of their momentary emotional states. Deciphering the true momentary emotional state of an individual requires extra-linguistic context and world knowledge. Even then, one can be easily mistaken.

- The inferences we draw in this paper are based on aggregate trends across large populations. We do not draw conclusions about specific individuals or momentary emotional states.

## Acknowledgments

Thank you to Krishnapriya Vishnubhotla for the Emotion Dynamics code-base, which set the ground work for computing instance average emotion scores for instances. This research was supported by NSERC, Digital Research Alliance of Canada (alliancecan.ca), Alberta Innovates, and DeepMind. This research project is funded by the Bavarian Research Institute for Digital Transformation (bidt), an institute of the Bavarian Academy of Sciences and Humanities. The author is responsible for the content of this publication.

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

## APPENDIX

## A  LexO vs. ML Methods for Emotion Labelling

In Table 5 we show the characteristics of the Lexicon-Only (LexO) and Machine Learning (ML) based methods for emotion labelling instances.

## B  Impact of Selectively Using the Lexicon

The continuously labeled emotion lexicons include words that may be very mildly associated with an emotion category or dimension. It is possible that the very low emotion association entries may in fact mislead the system, resulting in poor emotion arcs. We therefore also investigated the quality of emotion arcs by generating them only from terms with an emotion association score greater than a pre-chosen threshold; thereby using the lexicon entries more selectively. We systematically varied the threshold to study what patterns of threshold lead to better arcs across the emotion test datasets.[11]

Overall, we observed that valence benefits from including all terms, even lowly associated emotion words, as the optimal threshold across continuous and categorically datasets is 0 with a few notable exceptions (SemEval 2014 LiveJournal, SemEval 2014 tweets, and V-OC). Generally, including only terms with emotion scores above 0.33 to 0.5 improves the quality of emotion arcs.

## C  Impact of the Quality of Instance-Level Emotion Labeling on Emotion Arcs

Figure 5 shows the results for the Oracle System on the categorically labeled SemEval 2014 datasets. We observe similar patterns as discussed in the paper for the SemEval 2018 V-OC dataset. (Note that the instance-level random-guess baseline is dependent on the number of class labels; thus, the minimum Oracle System Accuracy at which positive correlations with gold arcs appear is different across the datasets.)

---

[11]Note that our goal is not to determine the predictive power of the system on new unseen test data. For that one would have to determine thresholds from a development set, and apply the model on unseen test data.

| Method | Pros | Cons |
|---|---|---|
| LexO | • *Interpretable*: One can easily examine the words in the text that are driving the emotion arc scores.

• *Simple and Accessible*: Does not require extensive infrastructure or programming expertise to execute the method.

• *Efficient & Environmentally Friendly*: Does not require high compute power.

• *Widely Applicable*: Does not require domain specific training data.

• *Captures Context by Aggregation:* As more data is available per bin, the predicted emotion arc gets closer to the true emotion arc because with more data it is more likely that: If bin $x$ is expressing more of an emotion than bin $y$, then $x$ has higher average word–emotion association score than $y$. | • *Poor at Instance Level*: Often not very accurate at labelling emotions of individual instances (sentences, tweets, etc.). Does not take context and long-distance dependencies into account.

• *Less Customization*: Does not usually capture domain-specific ways of expressing emotions; although, if available, domain-specific emotion lexicons can mitigate this limitation. |
| ML | • *Great at Instance Level*: Achieves state-of-the-art performance on emotion labelling instances.

• *Better at Capturing Context*: Considers context and long-range dependencies effectively. | • *Hard to interpret/explain the output.*

• *Requires extensive compute power and storage.*

• *Requires extensive computational expertise.*

• *Requires domain-specific training data.* |

Table 5: Characteristics of the Lexicon-Only (LexO) and Machine Learning (ML) based methods.

## D  African Languages

Information on the African languages used in our study are shown in Table 6.

## E  Emotion Labelled Datasets

In Table 7, we shown the emotion labelled instances for Arabic, and Spanish languages.

## F  Code and Resources

The code and resources used are made freely available on the project homepage.[12] The code allows for easy generation of high-quality emotion arcs for the provided text stream (especially useful for those outside of computer science).

---

[12]https://github.com/dteodore/EmotionArcs

| Test data | Emotion | Instance OS: Accuracy | Bin Size | | | | | |
|---|---|---|---|---|---|---|---|---|
| | | | 1 | 10 | 50 | 100 | 200 | 300 |
| SemEval 2014 (LiveJournal) | valence | 0 | -0.474 | -0.853 | -0.935 | -0.959 | -0.974 | -0.994 |
| | | 20 | -0.154 | -0.451 | -0.750 | -0.850 | -0.943 | -0.978 |
| | | 40 | 0.113 | 0.348 | 0.649 | 0.780 | 0.824 | 0.864 |
| | | 60 | 0.378 | 0.839 | 0.947 | 0.958 | 0.981 | 0.995 |
| | | 80 | 0.683 | 0.933 | 0.947 | 0.964 | 0.985 | 0.997 |
| | | 100 | 1.000 | 1.000 | 1.000 | 1.000 | 1.000 | 1.000 |
| SemEval 2014 (SMS) | valence | 0 | -0.361 | -0.777 | -0.863 | -0.850 | -0.829 | -0.798 |
| | | 20 | -0.126 | -0.387 | -0.655 | -0.735 | -0.774 | -0.806 |
| | | 40 | 0.080 | 0.229 | 0.478 | 0.621 | 0.691 | 0.761 |
| | | 60 | 0.358 | 0.756 | 0.866 | 0.877 | 0.886 | 0.903 |
| | | 80 | 0.654 | 0.880 | 0.894 | 0.901 | 0.916 | 0.932 |
| | | 100 | 1.000 | 1.000 | 1.000 | 1.000 | 1.000 | 1.000 |
| SemEval 2014 (tweets sarcasm) | valence | 0 | -0.562 | -0.797 | -0.995 | | | |
| | | 20 | -0.160 | -0.652 | -0.887 | | | |
| | | 40 | 0.180 | 0.707 | 0.958 | | | |
| | | 60 | 0.502 | 0.904 | 0.995 | | | |
| | | 80 | 0.764 | 0.921 | 0.999 | | | |
| | | 100 | 1.000 | 1.000 | 1.000 | | | |
| SemEval 2014 (tweets) | valence | 0 | -0.403 | -0.823 | -0.917 | -0.919 | -0.921 | -0.923 |
| | | 20 | -0.170 | -0.480 | -0.780 | -0.870 | -0.908 | -0.911 |
| | | 40 | 0.092 | 0.281 | 0.553 | 0.694 | 0.824 | 0.882 |
| | | 60 | 0.367 | 0.783 | 0.913 | 0.916 | 0.917 | 0.918 |
| | | 80 | 0.663 | 0.914 | 0.918 | 0.918 | 0.920 | 0.922 |
| | | 100 | 1.000 | 1.000 | 1.000 | 1.000 | 1.000 | 1.000 |

Correlation 0.000 — 1.000

Figure 5: **Valence Arcs:** Spearman correlations of the arcs generated using the **sentiment-classification Oracle System** with the gold arcs. Note: 'OS (Accuracy)' refers to the accuracy of the Oracle System on instance-level sentiment classification.

| Language | Family | Spoken In | #Speakers |
|---|---|---|---|
| Amharic | Afroasiatic | Ethiopia | 57.5 million |
| Hausa | Afroasiatic | northern Nigeria, Ghana, Cameroon, Benin, Togo, Ivory Coast | 72 million |
| Igbo | Niger–Congo | Southeastern Nigeria | 30 million |
| Kinyarwanda | Niger–Congo | Rwanda | 9.8 million |
| Swahili | Bantu | Tanzania, Kenya, Mozambique | 200 million |
| Yoruba | Niger–Congo | Southwestern and Central Nigeria | 52 million |

Table 6: Indigenous African languages included in our study.

| Dataset | Source | Domain | Dimension | Label Type | #Instances |
|---|---|---|---|---|---|
| SemEval 2018 Ar (V-OC) | Mohammad et al. (2018) | tweets | valence | categorical (-3,-2,...3) | 1,800 |
| SemEval 2018 Ar (V-Reg) | Mohammad et al. (2018) | tweets | valence | continuous (0 to 1) | 1,800 |
| SemEval 2018 Es (V-OC) | Mohammad et al. (2018) | tweets | valence | categorical (-3,-2,...3) | 2,443 |
| SemEval 2018 Es (V-Reg) | Mohammad et al. (2018) | tweets | valence | continuous (0 to 1) | 2,443 |

Table 7: Key information pertaining to the **Arabic (Ar) and Spanish (Es) emotion-labeled tweets datasets** we use in our experiments. #Instances includes the train, development, and test sets for the SemEval 2018 Task 1.