# OpenReview forum: "Evaluating Emotion Arcs Across Languages: Bridging the Global Divide in Sentiment Analysis"
_EMNLP/2023/Conference — EMNLP 2023 Findings_

### Official Review · Reviewer_ZKef · 2023-07-28

**Soundness:** 3

**Excitement:**

3: Ambivalent: It has merits (e.g., it reports state-of-the-art results, the idea is nice), but there are key weaknesses (e.g., it describes incremental work), and it can significantly benefit from another round of revision. However, I won't object to accepting it if my co-reviewers champion it.

**Paper Topic And Main Contributions:**

This paper addresses the evaluation and generation of emotion arcs, which represent how an individual or population's emotions change over time. The study compares two common methods for generating emotion arcs: Lexicon-Only (LexO) and Machine-Learning (ML) approaches. The paper systematically evaluates automatically generated emotion arcs using 42 diverse datasets in 9 languages. The authors also experimented with automatic translations of English emotion lexicons to generate emotion arcs in less-resourced languages.

The main contributions are:

- Evaluation of automatically generated emotion arcs.
- Comparison of two common methods for generating emotion arcs: LexO and ML approaches.
- Experiments on 42 diverse datasets in 9 languages, including indigenous African languages, Arabic, and Spanish.
- Authors claim they will release the data and code for easy generation of high-quality emotion arcs.

**Questions For The Authors:**

What is the influence of the methods across the languages and domains?

What is the motivation of choosing the specific metrics used? Have you considered others?

About translation, have you manually validated the accuracy and appropriateness of the translated emotion lexicons (sample size) for different languages?

Further analysis of the impact of OOV terms on emotion arcs could enhance the paper's insights.

What are the main limitations of the method used?

**Reasons To Accept:**


1. Original Contribution: The paper presents novel research on evaluating emotion arcs, a topic that has received limited attention in previous work.

2. Comparative Analysis: The paper compares two common methods for generating emotion arcs (LexO and ML) and highlights the strengths of the LexO approach in generating high-quality emotion arcs.

3. Evaluation: The paper conducts a systematic and quantitative evaluation of automatically generated emotion arcs using a diverse set of datasets in multiple languages.

**Reasons To Reject:**

Weak Presentation and Clarity: The paper's writing and organization may be unclear, making it difficult for readers to understand the research methods and findings. No examples are provided in the paper.

Evaluation of different datasets: The authors assert that they conducted experiments with diverse datasets; however, the paper does not present all of the results obtained from these experiments.

Evaluation Metrics: Although the paper uses correlation and RMSE values for evaluating emotion arcs, more discussion and analysis of the significance of these metrics in the context of emotion arcs would strengthen the research.

No quality analysis: The paper fails to provide a thorough quality analysis of the experiments, leaving a gap in assessing the reliability and robustness of the results.

Limited Discussion on Language Variations: The paper briefly mentions variations in translation quality across different languages.  Emotions are culturally and linguistically influenced, and direct translations of emotion lexicons may not fully capture the intricacies and subtleties of emotions in different languages and cultures.

No analysis of the impact of the OOV terms is provided.

The reproducibility of the results could be challenging. Not enough details are provided.

**Reproducibility:**

2: Would be hard pressed to reproduce the results. The contribution depends on data that are simply not available outside the author's institution or consortium; not enough details are provided.

**Reviewer Confidence:**

4: Quite sure. I tried to check the important points carefully. It's unlikely, though conceivable, that I missed something that should affect my ratings.

**Typos Grammar Style And Presentation Improvements:**

To improve the presentation of the paper, the authors should consider organizing the context more clearly to provide a better overview of the research. Additionally, they can include illustrative examples to guide readers through the steps of the methodology, making it easier for them to understand and follow the experiments.

---

> ### Author Rebuttal · Authors · 2023-08-28
>
> Dear reviewer,
>
> Thank you for your thoughtful comments on the presentation of the work. We have addressed these already in a revised version:
>
> – Added examples, qualitative analysis, limitations of translations: Since this is not a sentence-level task but rather a task about emotion arcs from thousands of sentences/tweets, it is difficult to provide examples of data behind emotion arcs in a paper (for space) and also challenging for qualitative analysis (due to the size of data to be assessed). However, we will make the data available on the project website. And we now include snippets from a few peaks and troughs of the arcs to give information about the context in those sets of sentences/tweets. This also enables qualitative analysis, as now we can explore where the predicted arc is farther away from the gold arc and examine the text in those areas. We will add the qualitative analysis and discussion of where translation quality impacts results in the revised version.
>
> – Experiments with many diverse datasets: We performed a vast number of experiments across diverse domains in 9 languages for 5 emotions, where instances had both categorical and continuous emotion scores. We systematically generate and evaluate emotion arcs on 42 datasets. We consider each dataset that has been labelled with a different emotion, emotion score type (categorical vs continuous), or in a different language as a separate dataset. As an overview, these included movie reviews with continuous and categorical emotion scores (2 datasets); SemEval 2014 data consisting of 4 domains (4 datasets); SemEval 2018 data in English, Arabic, and Spanish for valence, anger, fear, joy, sadness with both continuous and categorical emotion scores (30 datasets); and SemEval 2013 data for 6 Indigenous African languages (6 datasets). Due to the limited space, we show results in the main paper and appendix for 28 datasets. This means that 14 datasets have been left out due to space however, all result tables are also available on our project website. Generally, there were similar trends across datasets and languages, where increased bin size improved performance.
>
> – Impact of OOV terms: We performed experiments where words not found in the lexicon were either assigned a neutral score such as 0 (‘assigned neutral score’), or were disregarded (‘assigned NA’). In Figure 1 (pg. 4) we show the results on the quality of emotion arcs when using these two methods. At smaller bin sizes it was found the OOV method is slightly more impactful and ‘assigned NA’ generally performed better. However, at larger bin sizes (e.g., 200-300) both methods performed better than the other an approximately equal number of times. This is described further in Section 4 (lines 268, 288, 304). In Figure 2, for simplicity we only show the results for handling OOV as ‘assigned NA’ since both methods were comparable and this method performed better more often than not (lines 423-425).
>
> Q1. What is the influence of the methods across the languages and domains?
>
> The main things that influence results for lexicon-based methods for any language or domain are:
> – Quality and coverage of lexicon for the terms in the test set: the higher the better
> – Size of bins (aggregation): the higher the better
>
> The main things that influence results for ML methods for any language or domain are:
> – Quality, size, and domain match of labeled training data with test data: the higher the better
>
> Q2. What is the motivation of choosing the specific metrics used? Have you considered others?
>
> Spearman correlation and RMSE are the most commonly used metrics to determine the closeness of two time series arcs. Spearman correlation determines whether the shapes of the arcs are similar, without regard to the exact values. Thus this metric is ideal for most applications where the goal is to determine shape accurately (where the need is to determine whether emotions increased since the last comparison point or decreased). RMSE requires the shape and values to be close. Mathematically, RMSE is the standard deviation of the residuals. Residuals represent the distance between the regression line and the data points. Examining results with both metrics gives us a better picture of arc quality in different application scenarios.
>
> We also considered KL Divergence, but that metric is asymmetric and is more suitable for probability distributions.
> Pearson correlation largely produced the same results as Spearman.
>
> Q3. About translation, have you manually validated the accuracy and appropriateness of the translated emotion lexicons (sample size) for different languages?
>
> There are several works that have examined the quality of Google Translate’s translations over the last decade for various language pairs. The resource is also highly used throughout the world. We did a systematic study for some language pairs in our past work and found the suitability of translations to be high (> 90%).
>
> Q4. Further analysis of the impact of OOV terms on emotion arcs could enhance the paper's insights.
>
> We addressed this above, however we summarize the key takeaways. At smaller bin sizes it was found the OOV method is slightly more impactful and ‘assigned NA’ generally performed better. However, at larger bin sizes (e.g., 200-300) both methods performed better than the other an approximately equal number of times. This is described further in Section 4 (lines 268, 288, 304).
>
> Q5. What are the main limitations of the method used?
>
> Section 9 (Limitations) in the paper provides the limitations.
>
> Re Illustrative example:
> With the extra page available if the paper is accepted, we will add a figure showing the steps in the construction of the emotion arcs. We now also added a bulleted list of steps for the methodology of creating emotion arcs in the main text.
>
> Once again, thank you for your time and for engaging with the material in this paper and our author response. We see the time and thought you have put into this and very much appreciate it.

---

### Official Review · Reviewer_MGk7 · 2023-08-05

**Soundness:** 3

**Excitement:**

2: Mediocre: This paper makes marginal contributions (vs non-contemporaneous work), so I would rather not see it in the conference.

**Paper Topic And Main Contributions:**

The purpose of this paper is to evaluate the advantages of automatically generating emotion arcs by exploring lexical-based and machine learning approaches. To achieve that, well-established emotion lexicons are employed to construct emotion arcs using diverse data sources, including tweets, reviews, sentences, and various languages. Subsequently, the constructed emotion arcs are assessed using the two approaches utilized in this study.

**Reasons To Accept:**

- This study focuses on the assessment of a challenging task across multiple data sources and languages.

**Reasons To Reject:**

- The description of the evaluation and construction of emotion arcs in this paper lacks clarity. It would have been beneficial to include a diagram illustrating the process of constructing emotion arcs from tweets and reviews. Additionally, the justification for using different bin sizes is insufficient. This may be due to the chosen datasets in this paper, which may not fully capture the nuances of storytelling and playwriting commonly found in narratives and stories.

- In this paper, there is a discussion about using different levels of accuracy (line 322) when evaluating neural network models. However, it is unclear why this choice is important or how it contributes to the creation of emotion arcs. Providing further justification would be beneficial in helping readers understand which level of accuracy is most suitable for generating emotion arcs, such as high, medium, or low levels of accuracy.

- There appears to be a lack of experiments in this paper, considering the authors' claim in line 13 that they conducted experiments on 42 diverse datasets. However, only a portion of these datasets is presented in the paper. Furthermore, the experiments included in the paper fail to provide sufficient justification for the quality of the emotion arcs generated by both the lexical-based and machine learning approaches. It is necessary to conduct additional experiments and analyses to demonstrate the strengths and weaknesses of the two approaches employed in generating emotion arcs.

-----------------------------------------------
I thank the authors for their response to the comments.

**Reproducibility:**

2: Would be hard pressed to reproduce the results. The contribution depends on data that are simply not available outside the author's institution or consortium; not enough details are provided.

**Reviewer Confidence:**

3: Pretty sure, but there's a chance I missed something. Although I have a good feel for this area in general, I did not carefully check the paper's details, e.g., the math, experimental design, or novelty.

---

> ### Author Rebuttal · Authors · 2023-08-28
>
> Dear Reviewer,
>
> Thank you for your thoughtful comments and helping us to improve the description and presentation of the work.
>
> With the extra page available if the paper is accepted, we will add a figure showing the steps in the construction of the emotion arcs. We now also added a bulleted list of steps in the main text.
>
> Bin Sizes: Choice of bin size depends on the application and available data. For example, if a company wants to know how the proportion of angry comments has changed from month to month in the last 10 years, then the bin size to use is month. If instead, they want to know how things changed on a day by day basis over the last 45 days, then the bin size to use is a day. Depending on the volume of relevant data in the bin chosen, the bins may include a larger or smaller number of instances. We did experiments with many different amounts of data per bin to show that using bins with 100+ instances leads to better results. The systematic study and the results show how the quality of the arc changes with bin size so that practitioners can use suitable bin sizes for their particular data and task. Overall, smaller bin sizes provide higher granularity, but may not always have a sufficient amount of data for accurate arcs; therefore, using bigger bins is then preferable.
>
> Neural models: This paper additionally describes results with neural arcs to determine whether they provide markedly higher quality arcs than lexicon-based methods and under what conditions. We show that in most cases the improvement is only slight. The neural models are markedly better however when the bin size is very small (approaching 1 or few instances). However, in most applications of emotion arcs, especially those involving social media data, one has access to thousands, if not, tens or hundreds of thousands of instances, and so lexicon methods with bin sizes of a few hundred provide high quality arcs with little room left for improvement using more complex methods.
>
> Experiments with many diverse datasets: We performed a vast number of experiments across diverse domains in 9 languages for 5 emotions, where instances had both categorical and continuous emotion scores. We systematically generate and evaluate emotion arcs on 42 datasets. We consider each dataset that has been labelled with a different emotion, emotion score type (categorical vs continuous), or in a different language as a separate dataset. As an overview, these included movie reviews with continuous and categorical emotion scores (2 datasets); SemEval 2014 data consisting of 4 domains (4 datasets); SemEval 2018 data in English, Arabic, and Spanish for valence, anger, fear, joy, sadness with both continuous and categorical emotion scores (30 datasets); and SemEval 2013 data for 6 Indigenous African languages (6 datasets). Due to the limited space, we show results in the main paper and appendix for 28 datasets. This means that 14 datasets have been left out due to space however, all result tables are also available on our project website. Generally, there were similar trends across datasets and languages, where increased bin size improved performance.
>
> Once again, thank you for your time and for engaging with the material in this paper and our author response. We see the time and thought you have put into this and very much appreciate it.

---

### Official Review · Reviewer_QjH9 · 2023-08-09

**Soundness:** 4

**Excitement:**

4: Strong: This paper deepens the understanding of some phenomenon or lowers the barriers to an existing research direction.

**Missing References:**

Öhman, E., 2021, December. The validity of lexicon-based sentiment analysis in interdisciplinary research. In Proceedings of the Workshop on Natural Language Processing for Digital Humanities (pp. 7-12).

Öhman, E. and Rossi, R., 2023. Affect as a proxy for literary mood. Journal of Data Mining and Digital Humanities: NLP4DH.

Although Matthew Jockers' work on the syuzhet package has not been used much in academic research, it is still perhaps the most discussed emotion arc generator in DH and perhaps should be mentionded in this paper? At the very least perhaps:
Gao, J., Jockers, M.L., Laudun, J. and Tangherlini, T., 2016, November. A multiscale theory for the dynamical evolution of sentiment in novels. In 2016 International Conference on Behavioral, Economic and Socio-cultural Computing (BESC) (pp. 1-4). IEEE.


**Paper Topic And Main Contributions:**

This paper contributes to the ongong discussion and commparison between lexicon- and machine learning-based methods in emotion research. Additionally, it provides important cross-linguistic resources and new methods in implementing emotion detection methods for lower-resourced languages.


**Questions For The Authors:**

In section 4 the authors discuss the different approaches to emotion labeling and how the annotation guidelines are not consistent. The authors' solution is to us a predeterming emotion-association threshold. However, some recent papers by Ohman et al. (2021&2023 in particular) have shown that emotion intensity lexicons work better for real-world congruent results than lexicons than simply list associations. Did the authors consider using emotion intensity over simple association?

**Reasons To Accept:**

The authors convincingly present further evidence that lexicon-based methods have their place in computational sentiment analysis and emotion detection in particular and show that in applied task where usefulness is more important than accuracy metrics lexicon-based methods often outperform purely data-driven methods.

It is unfortunate that many researchers in emotioion detection are forced by a review process slanted towards achieving higher accuracy metrics to adapt machine learning methods in order to get their paper published. The recent influx of papers showing that these state-of-the-art approaches are not necessarily the most appropriate for applied downstream tasks are extremely important to enable a more accurate overviews of the emotional content of textual data in particular.

**Reasons To Reject:**

This paper has no weaknesses to mention.

**Reproducibility:**

4: Could mostly reproduce the results, but there may be some variation because of sample variance or minor variations in their interpretation of the protocol or method.

**Reviewer Confidence:**

4: Quite sure. I tried to check the important points carefully. It's unlikely, though conceivable, that I missed something that should affect my ratings.

**Typos Grammar Style And Presentation Improvements:**

There are no issues with language or presentations. This is a complete and polished paper.

---

> ### Author Rebuttal · Authors · 2023-08-28
>
> Dear reviewer,
>
> Thank you for your thoughtful and kind review. Thank you also for providing the valuable references. We have incorporated them in the revised version. Also, as you are suggesting, emotion intensity lexicons should be explored in addition to association lexicons since they contain more granular information. We use both association lexicons which we refer to as categorical lexicons scores, and emotion intensity lexicons which we refer to as continuous lexicon scores. The emotion intensity lexicon we use for anger, fear, joy, and sadness is the NRC Emotion Intensity Lexicon (continuous scores from 0 to 1) and for valence, the NRC Valence, Arousal, Dominance Lexicon which has scores from 0 to 1. In the result tables we use the column ‘Lexicon Scores’ to refer to whether an association or emotion intensity lexicon is being used. In the scenario the ‘Lexicon Scores’ column is not included in the table for space (this occurs once in Figure 4), we use the emotion intensity lexicon. We’ll make sure this is clear in the table caption as well, thank you for pointing this out. Indeed, as you say and like the paper you pointed to “The validity of lexicon-based sentiment analysis in interdisciplinary research” does very well, we hope this paper also encourages more thoughtfulness in recognizing the advantages and disadvantages of different methods, rather than simply focusing SOTA results. We see the time and thought you have put into this and very much appreciate it.

---

### Meta-Review · Area_Chair_fqns · 2023-09-19

**Recommendation:** 3

**Metareview:**

The paper addresses emotion arcs using lexicon-based and machine learning approaches. The reviewers have praised the paper's originality addressing a topic that was largely unexplored in previous work.

While one of the reviewers is very positive about the paper, the other two reviewers have identified several points that need improvement. For example, they have pointed to the lack of clarity in the evaluation and the description of experiments.

---

### Decision · Program_Chairs · 2023-10-07

**Decision:**

Accept-Findings

**Comment:**

The paper addresses emotion arcs using lexicon-based and machine learning approaches. The reviewers have praised the paper's originality addressing a topic that was largely unexplored in previous work.

While one of the reviewers is very positive about the paper, the other two reviewers have identified several points that need improvement. For example, they have pointed to the lack of clarity in the evaluation and the description of experiments.